# Melatonin Attenuates Spinal Cord Injury in Mice by Activating the Nrf2/ARE Signaling Pathway to Inhibit the NLRP3 Inflammasome

**DOI:** 10.3390/cells11182809

**Published:** 2022-09-08

**Authors:** Haoyu Wang, Haifan Wang, Heng Huang, Zhigang Qu, Dong Ma, Xiaoqian Dang, Quanyu Dong

**Affiliations:** 1Department of Orthopedics, The Second Affiliated Hospital of Xian Jiaotong University, Xi’an 710004, China; 2Department of Hand and Foot Surgery, The Affiliated Hospital of Qingdao University, Shandong 266001, China

**Keywords:** melatonin, NLRP3 inflammasome, inflammation, spinal cord injury, Nrf2/ARE

## Abstract

Background: Spinal cord injury (SCI) is a central nervous system (CNS) trauma involving inflammation and oxidative stress, which play important roles in this trauma’s pathogenesis. Therefore, controlling inflammation is an effective strategy for SCI treatment. As a hormone, melatonin is capable of producing antioxidation and anti-inflammation effects. In the meantime, it also causes a neuroprotective effect in various neurological diseases. Nrf2/ARE/NLRP3 is a well-known pathway in anti-inflammation and antioxidation, and Nrf2 can be positively regulated by melatonin. However, how melatonin regulates inflammation during SCI is poorly explored. Therefore, it was investigated in this study whether melatonin can inhibit the NLRP3 inflammasome through the Nrf2/ARE signaling pathway in a mouse SCI model. Methods: A model of SCI was established in C57BL/6 mice and PC12 cells. The motor function of mice was detected by performing an open field test, and Nissl staining and terminal deoxynucleotidyl transferase dUTP nick end labeling were carried out to evaluate the survival of neurons. Mitochondrial dysfunction was detected by transmission electron microscopy (TEM) and by assessing the mitochondrial membrane potential. In addition, the expression of NLRP3 inflammasome and oxidative-stress-related proteins were detected through Western blot and immunofluorescence double staining. Results: By inhibiting neuroinflammation and reducing neuronal death, melatonin promotes the recovery of neuromotor function. Besides this, melatonin is able to reduce the damage that causes neuronal mitochondrial dysfunction, reduce the level of reactive oxygen species (ROS) and malondialdehyde, and enhance the activity of superoxide dismutase and the production of glutathione peroxidase. Mechanically, melatonin inhibits the activation of NLRP3 inflammasomes and reduces the secretion of pro-inflammatory factors through the Nrf2/ARE signaling. Conclusions: In conclusion, melatonin inhibits the NLRP3 inflammasome through stimulation of the Nrf2/ARE pathway, thereby suppressing neuroinflammation, reducing mitochondrial dysfunction, and improving the recovery of nerve function after SCI.

## 1. Introduction

The trauma of SCI is often associated with many complications [1,2], and its intricate pathogenesis makes it intractable. The severe primary injury and subsequent secondary injury, including mitochondrial dysfunction, inflammation, oxidative stress, etc., bring about nerve cell disorders and changes in the internal environment, eventually leading to neuronal apoptosis [3]. However, precisely how neuroinflammation affects pathological and physiological changes in spinal cord injuries is still unknown [1,4].

The NLRP3 inflammasome is essential in the development of CNS inflammation [5]. As a multiprotein complex assembled by intracytoplasmic pattern recognition receptors (PRRs), it is essential for the natural immune system [6,7]. Inflammasomes recognize pathogen-related molecular patterns (PAMPs) or host-derived danger signal molecules (DAMPs), then respond to external stimuli and activate the pro-inflammatory protease caspase-1. Then, caspase-1 cleaves the pro-IL-1β and pro-IL-18 into a mature form [8,9]. This procedure induces a larger inflammatory cascade, accelerates the inflammatory response, and increases pyroptosis. The activated inflammasomes after SCI can induce neuronal cell dysfunction and even apoptosis [10,11]. 

Melatonin (MT), a well-known hormone, can delay aging and improve sleep; it is synthesized mainly in the pineal gland [12,13]. It possesses various physiological functions, including scavenging oxygen free radicals, antioxidation, inhibiting lipid peroxidation, preventing DNA damage, and reducing the content of oxides in the body [14]. It has been shown that anti-inflammation, antioxidation, and inhibition of neuronal apoptosis enable melatonin to play a neuroprotective role [15]. Besides this, as an antioxidant, it can target mitochondria to alleviate their dysfunctions in many free radical-related diseases [16,17,18]. Although some researchers have reported that melatonin has a certain effect on neuroprotection and can alleviate SCI in rats by inhibiting inflammation [17,19,20], whether and how melatonin inhibits the NLRP3 inflammasomes and ROS in neurons after SCIhas not been clarified. 

The Nrf2/ARE pathway’s anti-inflammation and antioxidant functions have been confirmed in many neurological diseases, and NLRP3 is a major downstream target of its anti-inflammation mechanism [21,22]. It has been reported that melatonin is a positive modulator of Nrf2 [23]. However, whether melatonin can activate the Nrf2/ARE pathway to improve recovery from SCI has not been explained.

As mentioned above, mitochondrial dysfunction and inflammation are major pathological factors of SCI [1], while melatonin has been verified to target mitochondrial to reduce ROS levels in many diseases [16,17,18], can alleviate SCI by inhibiting inflammation [17,19,20], and can activate the Nrf2/ARE pathway, a well-known anti-inflammation and anti-oxidation signaling pathway [23,24,25]. We hypothesize melatonin alleviates SCI through Nrf2/ARE signaling to inhibit inflammation and oxidation.

In this study, to verify the function of melatonin in Nrf2/ARE/NLRP3 inflammasomes, an in vivo SCI model was constructed to observe melatonin’s effect on this trauma, and a corresponding in vitro model was constructed to clearly detect how it works on neuron cells. 

## 2. Materials and Methods

### 2.1. Animals and Study Design

C57BL/6 mice (half male, half female, 20–25 g, 8 weeks, n = 100) were purchased from Beijing Vitar River Laboratory Animal Technology Co. Ltd. (Beijing, China). All procedures were performed in accordance with the Guidelines for Laboratory Animal Care and Use issued by the National Institutes of Health of the United States. The animals were kept in a 12 h/12 h light/dark cycle at a standard temperature (22–24 °C) and freely provided food and water for one week to adapt to the environment before the experiment. All animals were randomly divided into four groups: The sham group (n = 25), SCI–vehicle group (n = 25), SCI+melatonin group (n = 25), and SCI+melatonin+ML385 (Nrf2 inhibitor) group (n = 25). As Figure 1B depicts, mice were sacrificed at the corresponding time points, 15 at 3 days after SCI (we chose this time point to observe how melatonin affects acute inflammation of SCI) and 10 at 28 days after SCI for each group.

### 2.2. Surgical Preparations

Firstly, the mice were deeply anesthetized with sodium pentobarbital (40 mg/kg); then, laminectomy was performed at the level of T9–T10 of the mouse spine, completely exposing the dorsal side of the spinal cord. Next, a mouse spinal cord impactor (RWD, Cat#Model Ⅲ, USA, 2 mm diameter, 12.5 g, 20 mm in height) was used to cause a moderate contusion on the surface of the spinal cord at T9–T10 (a more severe contusion often results in obvious mortality). After the operation, the surface of the spinal cord was hemorrhaged, and the mice had straight legs and drooping tails, which indicates that the mouse model of moderate SCI was constructed successfully. The muscles, fascia, and skin of the mice were sutured layer by layer and disinfected with iodophor. The mice that had suffered spinal cord injury were placed in an incubator until they woke up. To prevent urinary tract infections, the bladder was emptied manually every day for the next few days until the mice regained spontaneous urination function. The mice in the sham operation experienced the same procedure, excluding the spinal cord contusion.

Firstly, mice in the SCI+melatonin group and the SCI+melatonin+ML385 group were intraperitoneally injected with melatonin (M5250, Sigma, St. Louis, MI, USA), and in the latter case, ML385 (medchemexpress, Shanghai, China), at a dose of 30 mg/kg in 0.1 mL of the vehicle (DMSO and 0.9% NaCl, 1:3) once a day [14,26], 2 h after SCI, and injected on the following 2 days. The sham group and SCI–vehicle group were administered intraperitoneal injections of 0.1 mL of the vehicle (DMSO and 0.9% NaCl, 1:3) at the same frequency and times.

### 2.3. Behavioral Assessments

The motor function of each group of mice was evaluated by the Basso Mouse Scale (BMS) [26] at 1–28 days (the common period to reach plateau) [27] after spinal cord injury. The scoring ranged from 0 to 9 points, where 0 means that the mouse is completely paralyzed and 9 means that the mouse is completely normal. Three researchers who did not know the grouping information of the mice independently observed the mice for more than 5 min on open ground. Then, the range of motion, coordination of the limbs, positions of the claws, positions of hind limbs, and tail for each group of mice were scored, and the experiment was repeated three times. 

### 2.4. Tissue Preparation

At a specific time after SCI, mice were anesthetized as described before and then perfused with normal saline and 4% paraformaldehyde through the heart. Then, 1.5 cm of spinal cord tissue was taken from the injured site (0.75 cm above and below T9–T10) and fixed in a 30% sucrose solution for one week until the spinal cord tissue completely sank to the bottom of the container. Then, the tissue was embedded and fixed with an O.C.T compound (4583, SAKURA, Torrance, CA, USA), and frozen sections were formed. Finally, the spinal cord was cut into transverse and longitudinal sections with a thickness of 10 μm using a cryo sectioning machine (CM3050S, Leica, Heidelberg, Germany) for the next experiment. All sections were finally stored in a refrigerator at −20 °C.

### 2.5. Nissl Staining

First, we soaked the spinal cord slices in a 1:1 mixture of ethanol and chloroform for 24 h. Next, the slices were gradient dehydrated in 100% and 75% ethanol solution for 1 min, and slices were washed with deionized water 3 times, one minute at a time. Next, the spinal cord sections were stained with 0.5% cresol violet solution for 5–10 min and washed with distilled water. Then, the slices were differentiated in 75% ethanol for 15 min and dehydrated twice in 100% ethanol for 3 min each time. After 5 min of washing in xylene and fixing with a neutral resin glue, images of four random fields in the ventral horn of the spinal cord were collected using a light microscope (DMI4000B, Leica, Wetzlar, Germany), and the normal motoneurons were quantitatively analyzed.

### 2.6. TUNEL Staining

TUNEL staining was performed using an in situ apoptosis detection kit (Solarbio, China). Simply, at first, the frozen sections of the spinal cord three days after the spinal cord injury were rewarmed at room temperature for 2 h, and the spinal cord sections were incubated with a mixed solution of 0.3% Triton X-100 and 0.1% sodium citrate for 10 min. Then, they were rinsed with PBS 3 times, for 3 min each time. Next, the spinal cord slices were incubated with the TUNEL-mixed solution in a dark and humid environment at 37 °C for 1 h. After washing with PBS 3 times, they were incubated with DAPI solution for 15 min. Finally, images were taken using a fluorescence microscope (Olympus, Tokyo, Japan), and at the same time, the green spots in the blue nuclei were deemed apoptotic cells. The average total numbers of cells and apoptotic cells in 6 sections of each mouse were counted, and the ratio was calculated.

### 2.7. Transmission Electron Microscope Analysis

Three days after the SCI surgery, at which animals showed the highest levels of inflammation and ROS, mice were deeply anesthetized, and they were perfused with PBS and 2.5% glutaraldehyde through the heart until they became rigid. At the same time, 1 cm of the spinal cord tissue was cut from the spinal cord contusion point and immersed in a 2.5% glutaraldehyde solution for fixation and long-term storage. Next, the tissue was washed with PBS (3 min × 3 times). After the spinal cord tissue was fixed with acetic acid fixative (1%) for 1 h, it was washed with PBS 3 times and then dehydrated with an acetone solution gradient. After dehydration, the spinal cord tissue was incubated with a transdermal buffer prepared by mixing the embedding agent and acetone for 2 h. Next, they were embedded at 45 and 60 °C, 24 h at each temperature. Then, slices were processed using an ultra-thin microtome and stained with 1% uranyl acetate–lead citrate; mitochondria in the spinal cord of each group of mice were observed under a transmission electron microscope. Three or four discontinuous slices were taken at each spinal cord sample, three fields of view were selected, and the average of the measurements was taken.

### 2.8. Cell Culture and Treatment

The PC12 neuron cell line was purchased from the Cell Bank of the Chinese Academy of Sciences. It was cultured in Dulbecco’s modified Eagle’s medium (DMEM, Hyclone, UT, USA) supplemented with 10% FBS (Gibco, Grand Island, NY, USA) and 50 µg/mL penicillin/streptomycin (Gibco, Grand Island, NY, USA) at 37 °C with 5% CO_2_ and 95% air. After the cell state was stabilized, the cells were incubated with 20 μMol H_2_O_2_ and 5 mmol/L ATP for 6 h to induce an inflammatory state containing the expression of NLRP3 protein (this treatment will promote ROS’s production, then activates NLRP3 [28,29,30,31]), which was used in the later experiments of each group. Then, ML385 (6 μMol) and/or melatonin (60 μMol) [26] were incubated with the treated PC12 cells for 12 h. Then, the cell tissues of each group were collected for subsequent experiments.

### 2.9. Cell Viability

PC12 cells were seeded in a 96-well plate. A total of 5000 cells were seeded in each well, and they were cultured in an incubator containing 5% CO2 and 95% air at 37 °C for 24 h. After the cell state was stabilized, different doses of melatonin were added to different wells. After continuing the culture for 12 h, 25 μL of the CCK-8 solution and 175 μL of the fresh medium were added to each well, and these were co-cultured for 1 h at 37 °C. Then, the absorbance was measured at 450 nm using a microplate reader (Synery-2, BioTek, Winooski, VT, USA).

### 2.10. Determination of Intracellular ROS Accumulation

The fluorescent marker DCFH2-DA was used to measure the content of ROS in PC12 cells in each group, as described previously [26]. Firstly, PC12 cells were inoculated onto a 24-well plate, with 10^5^ cells in each well, and then cultured for 24 h; after the cell status was stable, the cells were treated with H_2_O_2_ (20 μMol) for 6 h under the action of the various treatment factors in the group. Then, we removed the medium and washed the cells with PBS 3 times, for 3 min each time. DCFH2-DA (10 μM) and 200 μL of DMEM mixed medium were incubated with each group of cells for 30 min. Then, a fluorescence microscope (Olympus, Tokyo, Japan) was used to collect images of each group of cells. 

### 2.11. JC-1 Staining

The PC12 cells were stained using a JC-1 staining kit (Solarbio, China). In short, firstly, 5 × 10^5^ cells were seeded in a 12-well plate, after the treatment of each group with the different treatment factors as described previously. A volume of 250 μL JC-1 fluorescent probe was added to each group of cells, and they were incubated together at 37 ℃ for 1 h. Next, the culture medium was removed, and cells were washed with buffer twice for 3 min each time. Finally, 500 μL of culture medium was added to the wells of each group, and imaging was performed using a fluorescence microscope (Olympus, Tokyo, Japan). Aggregates were observed in a specific red fluorescence excitation wavelength maximum at 585 nm and emission wavelength at 590; J-monomers were observed at the green fluorescence excitation wavelength maximum of 515 nm and emission wavelength at 529 nm. The red and green fluorescence intensities were analyzed by Image J software, and their ratio was taken as the relative mitochondrial membrane potential. 

### 2.12. Determination of Oxidative Stress

After the cells of each group were treated with different processing factors, various cell tissues were collected, and then a cell homogenate was prepared with cold phosphate-buffered saline. The activities of glutathione peroxidase (GSH-PX), malondialdehyde (MDA), and superoxide dismutase (SOD) were tested using GSH-PX, SOD, and MDA test kits (Solarbio, China) according to the manufacturer’s instructions.

### 2.13. Western Blot Analysis

The spinal cord tissue from the injury site and PC12 cells of each group were collected, and Western blotting was performed. The primary antibodies used in this experiment were as follows: Anti-Nrf2 (1:1000; Abcam, UK); anti-NQO-1 (1:1000; Abcam, UK); anti-HO-1 (1:1000; CST, USA); anti-NLRP3 (1:1000; Affinity, USA); anti-ASC (1:1000; Affitiny, USA); anti-Caspase-1 (1:1000; CST, USA); anti-IL-1β (1:1000; Abcam, UK); anti-GAPDH (1:1000; Abcam, UK); and anti-Lamin B1 (1:1000; Abcam, UK). After incubating the band with the primary antibody overnight, it was washed with TBST 3 times for 3 min each time; the band was then incubated with the corresponding secondary antibody for 2 h at room temperature. Then, the bands were detected by enhanced chemiluminescence reagent (ECL) (Thermo Fisher Scientific), and Image J software (Media Cybernetics, Georgia, MD, USA) was used for semi-quantitative analysis.

### 2.14. Immunofluorescence Analysis

The spinal cord slices 3 days after SCI in groups were rewarmed for 2 h at room temperature, and the PC12 cells treated with different factors in each group were fixed with paraformaldehyde. Those sections were incubated with 5% normal goat serum containing 0.3% tritonx-100 for 2 h. Then, they were washed with PBS 3 times, for 3 min each time. The primary antibody was incubated with spinal cord slices at 4 °C for 24 h. The primary antibodies used in the experiment were as follows: Anti-HO-1 (1:500; Abcam, UK); anti-IL-1β (1:500; Abcam, UK); anti-Nrf2 (1:1000; Abcam, UK); anti-NLRP3 (1:500; Affinity, USA); and anti-α-Tubulin (1:500; Abcam, UK). On the next day, the primary antibody was washed off with 1 × PBS, and the corresponding fluorescent secondary antibody (Alexa Fluor 594 goat anti-mouse IgG or Alexa Fluor 488 goat anti-rabbit IgG; 1:250, Thermo-Fisher Science) and the spinal tissues or cell tissues were subsequently incubated together for 2 h at room temperature. Next, the cell nuclei were stained with 4’6-diamino-2-phenylindole (DAPI) (Invitrogen, Carlsbad, CA, USA) solution. Finally, we used a fluorescence microscope (Olympus, Tokyo, Japan) to observe and image the spinal cord tissue, and we used ImageJ2x software to analyze the average optical density or count the fluorescence-positive cells in randomly selected images in different tissues.

### 2.15. Enzyme-Linked Immunosorbent Assay (ELISA)

The levels of IL-1β and IL-18 were measured in the PC12 cell culture medium using ELISA kits according to the manufacturer’s instructions (Elabscience, China).

### 2.16. Statistical Analysis

All the values are shown as the mean ± SEM. One-way ANOVA was used for multiple groups’ comparison, and Bonferroni’s post hoc test was performed if the variance was equal or a Kruskal–Wallis test if the variance was uneven. The BMS score was analyzed via two-way ANOVA with Tukey’s post hoc test. A value of *p* < 0.05 was considered statistically significant.

## 3. Results

### 3.1. Melatonin Promotes the Recovery of Motor Neurons after Spinal Cord Injury

To test melatonin’s effects on the motor function of mice that suffered SCI, first, the BMS score was calculated to assess the behavioral scores of mice within 28 days after SCI (Figure 1A). The score of mice that received sham operation was significantly better than the SCI–vehicle mice, indicating those mice’s motor function was severely damaged after spinal cord injury. On the 14th day after SCI, the damaged motor function was significantly improved, and this therapeutic effect was maintained until the 4th week. 

As Figure 1C,D show, motor neurons in the anterior horn of the spinal cord near the injury site were detected by Nissl staining. The motor neurons in the SCI–vehicle mice were significantly reduced compared with the mice that received a sham operation. At the same time, the motor neurons were significantly richer in mice treated with melatonin than in the SCI–vehicle mice, but this phenomenon was partly reversed by ML385.

Apoptosis of spinal cord cells was analyzed by means of TUNEL staining on spinal cord sections of mice in groups 3 days after the SCI surgery (Figure 1E,F). The programmed dead cells increased significantly after spinal cord injury. Compared with the SCI–vehicle mice, the mice that received MT presented a significantly reduced number of apoptosis-positive cells, but the ML385 treatment reversed this reduction.

We then observed the mitochondria in the spinal cord by transmission electron microscopy (Figure 1G,H). After SCI, the mitochondria in the spinal cord were destroyed, the vacuolation rate increased, and the chromatin was concentrated. With the comparison of the SCI model mice, the vacuoles of mitochondria in the mice treated with MT were few, and the cristae of the mitochondria also recovered. Meanwhile, the vacuole rate of mitochondria in the MT+ML385-treated mice was significantly higher than in the MT group, and the cristae of mitochondria were destroyed.

### 3.2. Melatonin Reduces Oxidative Damage to PC12 Cells Induced by H_2_O_2_

To clarify how melatonin influences cell oxidative damage, the oxidation products and mitochondrial membrane potential of PC12 cells in different treatment groups were analyzed in vitro. 

CCK-8 assay results showed that a low concentration of melatonin had a certain proliferation-promoting effect on PC12 cells (Figure 2A). However, when the concentration reached 80 µmol, melatonin significantly suppressed the viability of these cells. 

Then, we tested how melatonin affects the production of ROS in PC12 cells induced by H_2_O_2_. The results showed that melatonin significantly reduced the production of H_2_O_2_-induced ROS in PC12 cells (Figure 2B).

In the next test, we evaluated the mitochondrial membrane by JC-1 staining. It indicated that, compared with the H_2_O_2_ group, melatonin treatment significantly improved mitochondrial function and increased the mitochondrial membrane potential (Figure 2C,D). Meanwhile, compared with melatonin-treated mice, the mice that received both melatonin and ML385 showed reduced mitochondrial membrane potential. Then, we tested the contents of MDA and SOD and the activity of GSH-PX in PC12 cells. It showed that compared with the H_2_O_2_-treated mice, melatonin treatment reduced the content of MDA, increased the content of SOD, and increased the activity of GSH-PX simultaneously. However, this change was offset by ML385 (Figure 2E–G). 

Those results imply that melatonin can alleviate the oxidative damage to PC12 cells induced by hydrogen peroxide and protect mitochondrial function, by activating Nrf2/ARE signaling.

### 3.3. Melatonin Elevates the Level of Nrf2/ARE-Related Antioxidant Proteins and Inhibits the NLRP3 Inflammasome after SCIin Mice

The mechanism of melatonin’s neuroprotective effect was explored. Firstly, we detected antioxidant-related proteins and inflammation proteins in the spinal cords of mice 3 days after SCI using Western blotting. After melatonin treatment, the protein expression levels of Nrf2, HO-1, and NQO-1 were significantly increased (Figure 3A–D). Further, the expression of the NLRP3 inflammation-related protein in the SCI model mice was significantly higher than in the sham mice after the spinal cord injury surgery, and the melatonin-treated mice showed significantly reduced protein levels of NLRP3, ASC, caspase-1, and IL-1β compared to the SCI model mice (Figure 3E–I).

For further identification, the immunofluorescence double-staining technique was performed to test the protein level of HO-1 (Figure 4A,B) and IL-1β (Figure 4C,D) in the anterior horn motor neuron cells of each group’s mice. The results are consistent with those using Western blotting.

The above results indicate that melatonin has the potential to resist oxidative stress and inhibit NLRP3 inflammation after SCI in mice.

### 3.4. Melatonin Suppresses the NLRP3 Inflammasome through Activating the Nrf2/ARE Signalingy after SCI in Mice

As shown in Figure 5A,B, Nrf2 in mice that received MT treatment was significantly higher than in the SCI model mice. Meanwhile, with the Nrf2/ARE-pathway-specific inhibitor (ML385) treatment, the protein level of nuclear Nrf2 significantly decreased. Further, higher protein expression of HO-1 and NQO-1 were detected in the MT group compared with the SCI–vehicle group (Figure 5C–I). However, ML385 treatment partly counteracted this change. Interestingly, compared with SCI model mice, the NLRP3 inflammation-related protein in the melatonin-treated mice was significantly reduced, whereas it was significantly higher in the melatonnin+ML385-treated mice than in the mice that received the single melatonin treatment. 

Immunofluorescence double staining showed that Nrf2 in neuronal cells was significantly increased after melatonin treatment (Figure 6A,B), and the NLRP3 proteins in neuronal cells were significantly reduced. However, Nrf2 inhibitor ML385 treatment counteracted the reduction induced by melatonin. It suggests that melatonin can prevent the NLRP3 inflammasome via Nrf2/ARE signaling after SCI in mice.

### 3.5. Melatonin Suppresses the NLRP3 Inflammasome in PC12 Cells Treated with H_2_O_2_ via the Nrf2/ARE Defense Pathway

To explore the connection between the NLRP3 inflammasome and oxidative stress, we constructed an in vitro model in PC12 cells using H_2_O_2_ +ATP treatment. As shown in Figure 7A–B, the nuclear level of Nrf2 in the H_2_O_2_+MT-treated PC12 cells was significantly higher than in the model PC12 cells. ML385 was able to reverse this effect. As shown in Figure 7C–K, after melatonin treatment, the level of ARE antioxidant-related proteins in PC12 cells was significantly higher than in the model cells. Meanwhile, the level of NLRP3 inflammation-related proteins was significantly lower. Compared with the H_2_O_2_+MT-treated PC12 cells, the ARE antioxidant-related protein level was significantly reduced with additional ML385 treatment. On the contrary, the NLRP3 inflammation-related protein level was significantly increased. It can be seen that the suppression of melatonin on the NLRP3 inflammasome is closely associated with the Nrf2/ARE antioxidant pathway.

Furthermore, as shown in Figure 8A–D, we tested the level of Nrf2 and NLRP3 in PC12 cells by immunofluorescence double staining. It shows that, compared with that in the H_2_O_2_-treated PC12 cells, the rate of Nrf2-positive cells in the H_2_O_2_+MT group was significantly higher, and this increase was partly prevented by ML385 treatment. 

The expression of NLRP3 in PC12 cells that received different factors in each group showed exactly the opposite trend to the expression levels of Nrf2. Consistent with the in vivo results, the above results show that melatonin can suppress the NLRP3 inflammasome via Nrf2/ARE signaling in PC12 cells.

## 4. Discussion

SCI causes a serious strike to the nervous system, and consequent secondary injury makes it intractable. Severe neuroinflammation, oxidative stress, and mitochondrial dysfunction during secondary injury can cause huge damage to neurons after spinal cord injury [32,33]. Among these factors, inflammation is a promising therapeutic target for SCI [34]. As an endogenous hormone, melatonin has a physiological regulatory effect on the CNS. Recently, many researchers have proven that melatonin can also play an anti-inflammatory role by regulating a variety of cytokines [15,17]. Besides this, as an antioxidant, melatonin can reduce reactive oxygen species in tissues, making it appropriate for the treatment of diseases that involve oxidative injuries, such as SCI, cerebral ischemia-reperfusion, and Alzheimer’s disease [35,36]. In our present work, we established an SCI mouse model to explore the neuroprotective effect of melatonin on this trauma and its associated mechanisms. Through the BMS score, Nissl staining, and TUNEL staining, we proved that melatonin can speed up the recovery of motor function after SCI in mice. Further, we verified the protective effects of melatonin in both in vitro and vivo experiments, showing that they may derive from the inhibition of NLRP3 inflammation through stimulating Nrf2/ARE signaling. Furthermore, the in vitro study reveals neuronal cells to be a major target of melatonin in SCI. 

After SCI, many cells in the spinal cord are damaged, among which a large number of neuronal cells undergo programmed cell death and necrosis [37]. Neuronal cells, the main components of the CNS, not only transmit signals, but also secrete several neurotrophic factors and some chemical substances when stimulated by the outside world [38]. Spinal cord injury also stimulates the immune microenvironment, and various nerve cells in the CNS respond with immune responses to resist external interference [38,39]. Among them, neuroinflammatory and oxidative stress are essential [26]. The NLRP3 inflammasome, one important component of innate immunity in the CNS, after SCI [40], is activated by an exogenous stimulus, and then recruits and activates the related pro-inflammatory protease caspase-1. Activated caspase-1 promotes the maturation and secretion of interleukins and other related cytokines [10]. Increasing numbers of researchers have found that the NLRP3 inflammasome is vital for the cascade reaction after SCI [5]. Interestingly, our study suggested that the NLRP3 protein in the spinal cord tissue was expressed in large quantities after spinal cord injury, and related inflammatory proteins were also significantly upregulated. Compared with the SCI model mice and PC12 cells, melatonin significantly decreased NLRP3 inflammation-related proteins. This indicates that melatonin can effectively suppress NLRP3 inflammation after SCI.

Increased ROS is produced after SCI, which is considered to be a large factor in promoting the activation of NLRP3 inflammation [41,42,43]. The ROS in neurons mainly comes from the mitochondria. Once mitochondria are damaged, large amounts of ROS and oxidation products are produced, further aggravating the damage [44,45]. Meanwhile, it was also shown that the natural antioxidant, melatonin, can effectively prevent oxidative stress and reduce the production of ROS [16]. Fortunately, this phenomenon was repeated in our experiments. In our in vitro experiments, we found that more active oxygen was produced in PC12 cells treated with hydrogen peroxide. However, after they received melatonin treatment, the active oxygen significantly decreased. This hormone also elevated the content of SOD and the activity of GSH-PX, and it reduced the content of MDA. Further, by JC-1 staining and transmission electron microscopy observation, we found that melatonin can significantly increase the mitochondrial membrane potential and improve mitochondrial function. However, all these effects were reversed by ML385 treatment. This implies that melatonin inhibits oxidative damage and improves mitochondrial function by stimulating the Nrf2/ARE pathway.

Transcription factor Nrf2 is essential for regulating cellular oxidative stress [46]. It can induce the expression of certain antioxidant factors after activation, reduce the production of ROS, maintain the redox reaction, and keep cells in a stable state [46,47]. It has been shown that melatonin, as an activator of Nrf2, can inhibit NLRP3 inflammation, reduce atherosclerosis, and improve cerebral ischemia–reperfusion injury by activating Nrf2 [48,49]. As a result, Nrf2/ARE signaling is a potential target for inhibiting oxidative stress and improving spinal cord injuries [50]. In this research, we found that melatonin can promote the expression of Nrf2/ARE-related proteins after spinal cord injury. Similarly, in PC12 cells co-treated with H_2_O_2_ and ATP, we obtained the same conclusion after melatonin treatment. To further prove this, an Nrf2-specific inhibitor (ML385) was used in both the PC12 cells and mice spinal cord injury models. Interestingly, ML385 reversed the activation of the Nrf2/ARE pathway and the inhibition of NLRP3 inflammation. This suggests that melatonin inhibits NLRP3 inflammation after SCI via Nrf2/ARE signaling.

In general, our work proves that melatonin is a promising drug for spinal cord injury, and it provides a solid theoretical foundation for future clinical research. However, there are still some limitations. First of all, because of restricted conditions, we failed to use Nrf2-knockout mice to conduct in-depth research. Besides this, we should consider using different doses of melatonin in different periods of SCI to test how melatonin affects SCI more systematically and comprehensively. 

Clinically, our work in this study suggests the potential value of melatonin in the treatment of SCI to improve prognosis. It is worth mentioning that oral melatonin tablets are a common tonic with only limited side effects. Therefore, after more specific basic research, subsequent clinical trials on SCI patients would be safe.

## 5. Conclusions

In summary, after SCI, melatonin can inhibit oxidative stress and then alleviate the NLRP3 inflammasome via stimulating the Nrf2/ARE pathway. As a result, we suggest that melatonin can protect nerve cells and improve motor function after SCI. This provides a new therapeutic option for spinal cord injury.

## Figures and Tables

**Figure 1 cells-11-02809-f001:**
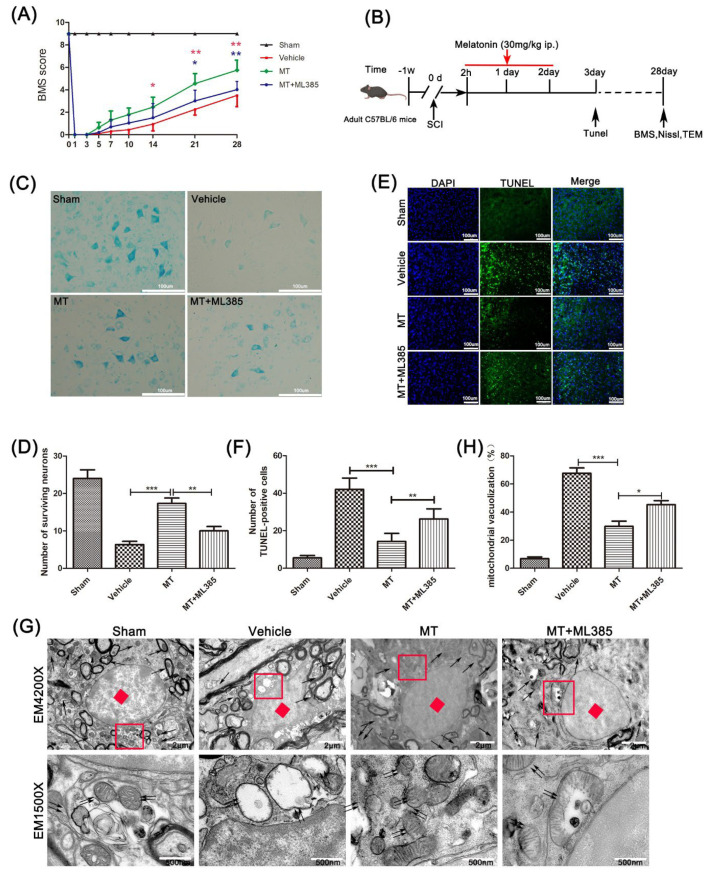
Melatonin can reduce neuron damage after SCI and speed up the recovery of motor function. (**A**) The BMS scores of mice four weeks after spinal cord injury to assess the motor function of mice in the sham, vehicle, MT, and MT+ML385 groups (n = 10, Tukey’s post hoc test followed by two-way ANOVA). (**B**) Design route of this research. (**C**,**D**) Nissl staining of the spinal cord anterior horn neurons in 4 groups 4 weeks after SCI (n = 6, scale bar = 100 µm); statistical analysis was used to evaluate survival. (**E**,**F**) TUNEL staining was used to evaluate the programmed cell death 3 days after SCI in each group (n = 6, scale bar = 100 µm). (**G**,**H**) TEM was used to observe the damage degree of mitochondria in the spinal cord tissue of mice and statistically analyze the vacuolation rate of mitochondria in the spinal cord of all 4 groups (n = 6, scale bar = 2µm; 500 nm). Data are shown as the means ± SD; * *p* < 0.05, ** *p* < 0.01, *** *p* < 0.001.

**Figure 2 cells-11-02809-f002:**
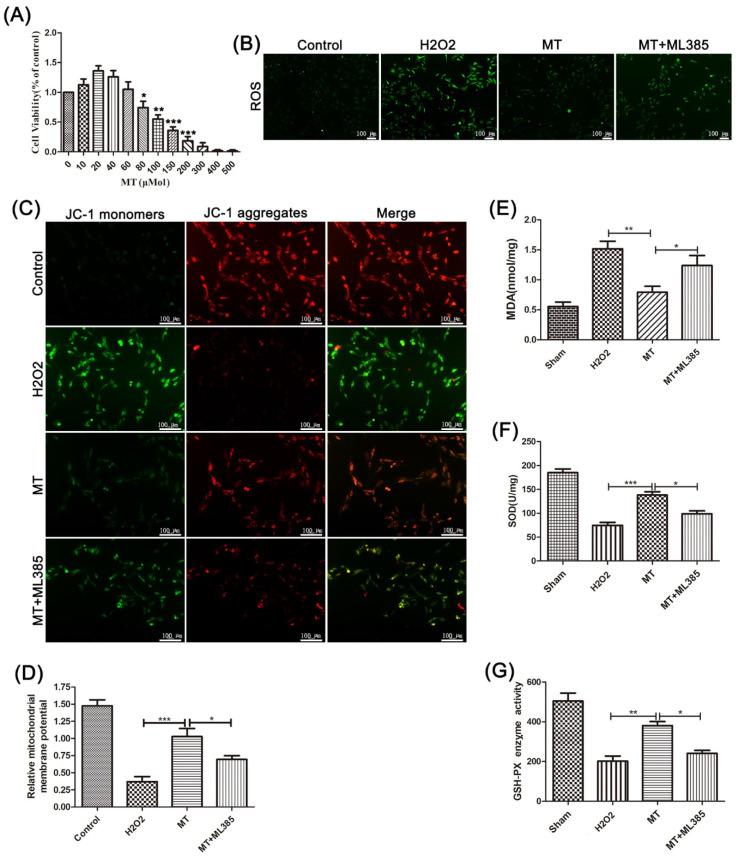
The oxidative damage scavenging effect of melatonin in neuron cells. (**A**) The survival rates of PC12 cells that received different concentrations of melatonin treatment were detected using a CCK-8 kit (n = 8). (**B**) The ROS produced in PC12 cells that received different treatments were detected by ROS staining (n = 6, scale bar = 100 µm). (**C**,**D**) The mitochondrial membrane potential in PC12 cells treated with drugs was detected by JC-1 staining (n = 8, scale bar = 100 µm). (**E**–**G**) The contents of SOD and MDA and the activity of GSH-PX were detected using the corresponding kits in PC12 cells treated with different treatment factors (n = 10). Data are shown as the means ± SD; * *p* < 0.05, ** *p* < 0.01, *** *p* < 0.001.

**Figure 3 cells-11-02809-f003:**
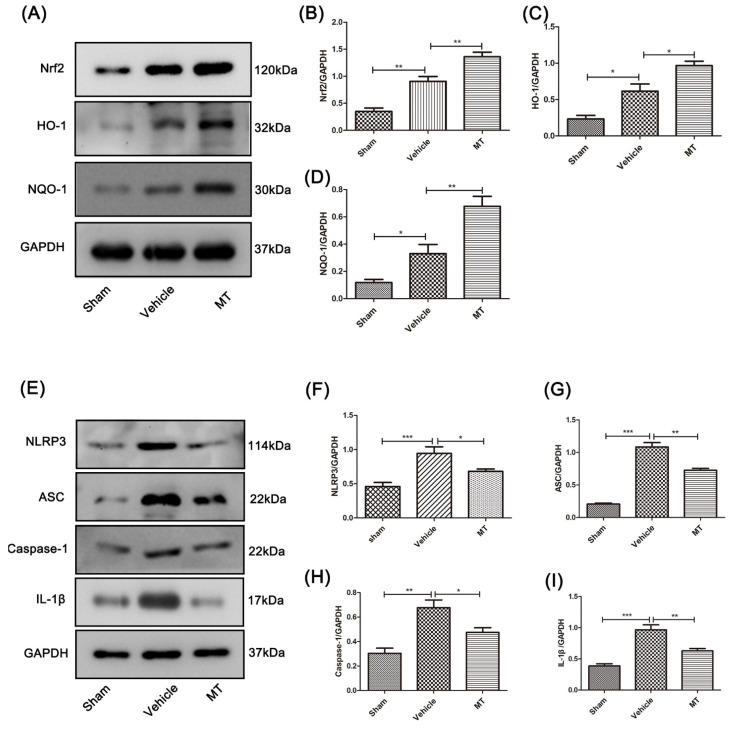
Melatonin can promote the activation of Nrf2/ARE-related proteins and inhibit the NLRP3 inflammasome in the spinal cord tissue near the injury site of mice 3 days after SCI. (**A**–**D**) The protein expression of Nrf2, HO-1, and NQO-1 in the spinal cord tissue near the injury site in each group was assessed by Western blotting (n = 6). (**E**–**I**) The protein expression of NLRP3, ASC, caspase-1, and IL-1β in the spinal cord tissue near the injury site was detected by Western blotting (n = 6). Data are shown as the means ± SD; * *p* < 0.05, ** *p* < 0.01, *** *p* < 0.001.

**Figure 4 cells-11-02809-f004:**
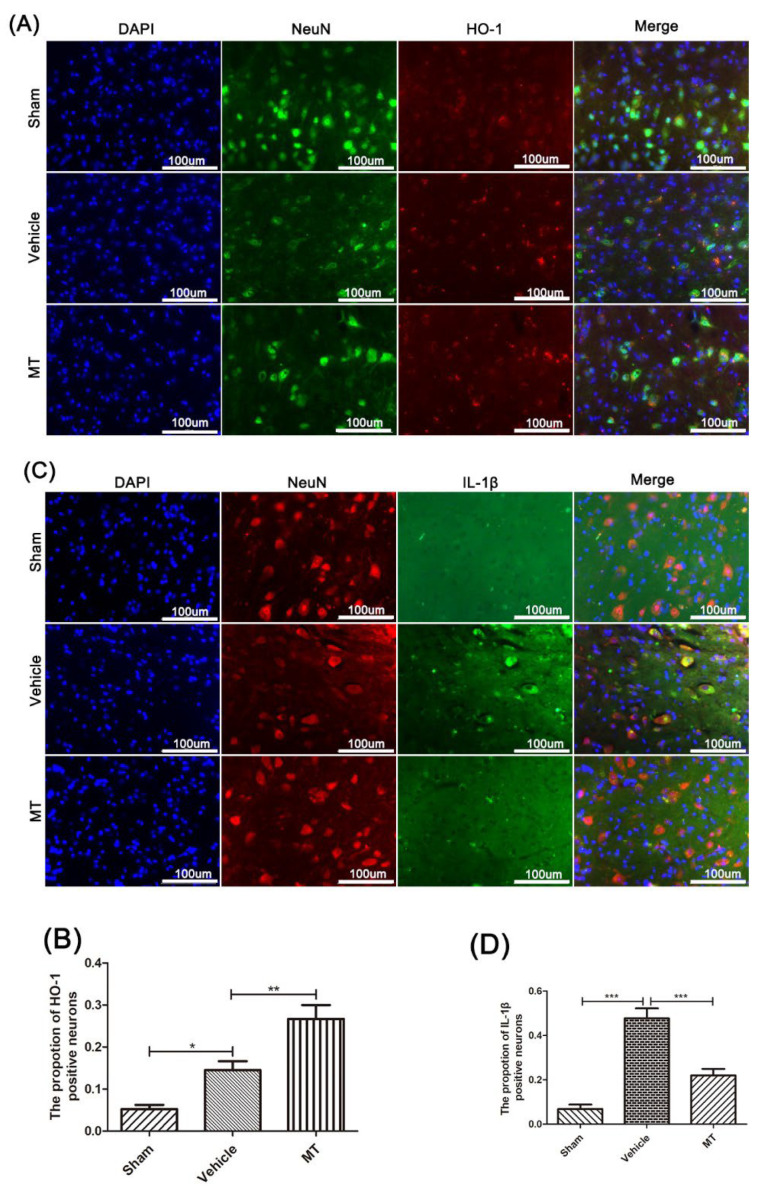
Immunofluorescence double staining of HO-1 and IL-1β in mouse anterior horn motor neuron cells near the injury site, 3 days after SCI. The protein expression of HO-1 (**A**,**B**) and IL-1β (**C**,**D**) was detected by the immunofluorescence double-staining technique, in anterior horn motor neuron cells near the injury site in each group’s mice (n = 6, scale bar = 100 µm). Data are shown as the means ± SD; * *p* < 0.05, ** *p* < 0.01, *** *p* < 0.001.

**Figure 5 cells-11-02809-f005:**
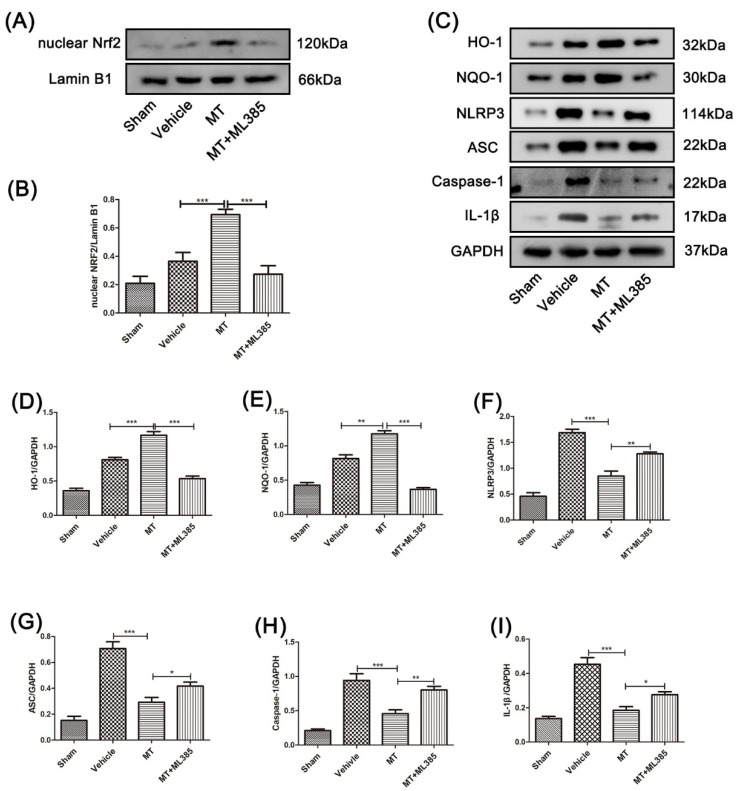
Melatonin can suppress the NLRP3 inflammasome in mice spinal cords near the injury site, 3 days after SCI, by activating the Nrf2/ARE pathway. (**A**,**B**) The nuclear Nrf2 in the spinal cord of mice in groups was detected by Western blotting (n = 8). (**C**–**I**) The levels of HO-1, NQO-1, NLRP3, ASC, caspase-1, and IL-1β in the spinal cord tissue of mice in groups were detected by Western blotting (n = 6). Data are shown as the means ± SD; * *p* < 0.05, ** *p* < 0.01, *** *p* < 0.001.

**Figure 6 cells-11-02809-f006:**
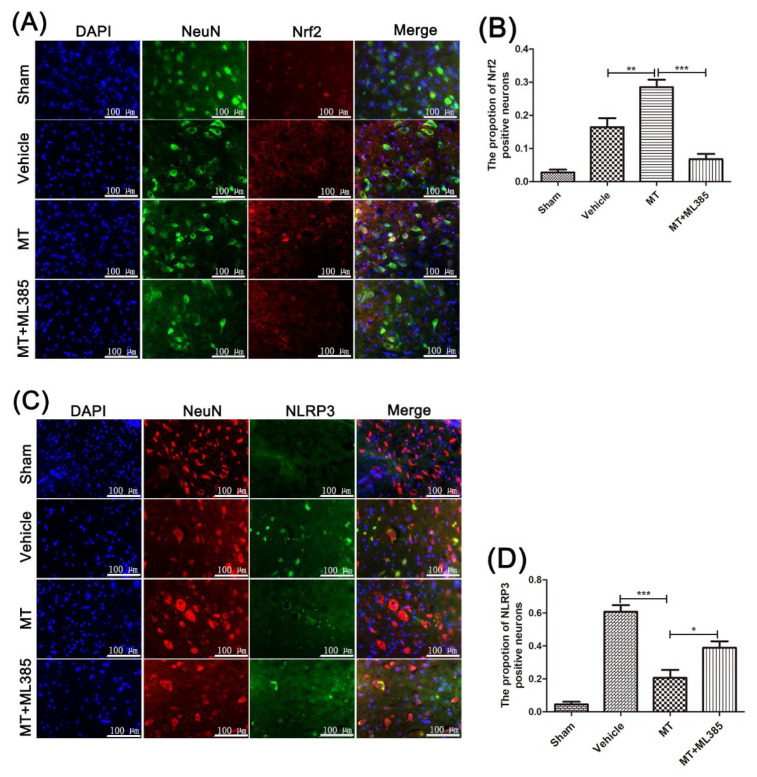
Immunofluorescence staining of Nrf2 and NLRP3 in spinal anterior horn motor neuron cells near the injury site of mice, 3 days after SCI. Nrf2 (**A**,**B**) and NLRP3 (**C**,**D**) proteins in the spinal anterior horn motor neuron cells of mice in groups were detected by immunofluorescence double staining (n = 6, scale bar = 100 µm). Data are shown as the means ± SD; * *p* < 0.05, ** *p* < 0.01, *** *p* < 0.001.

**Figure 7 cells-11-02809-f007:**
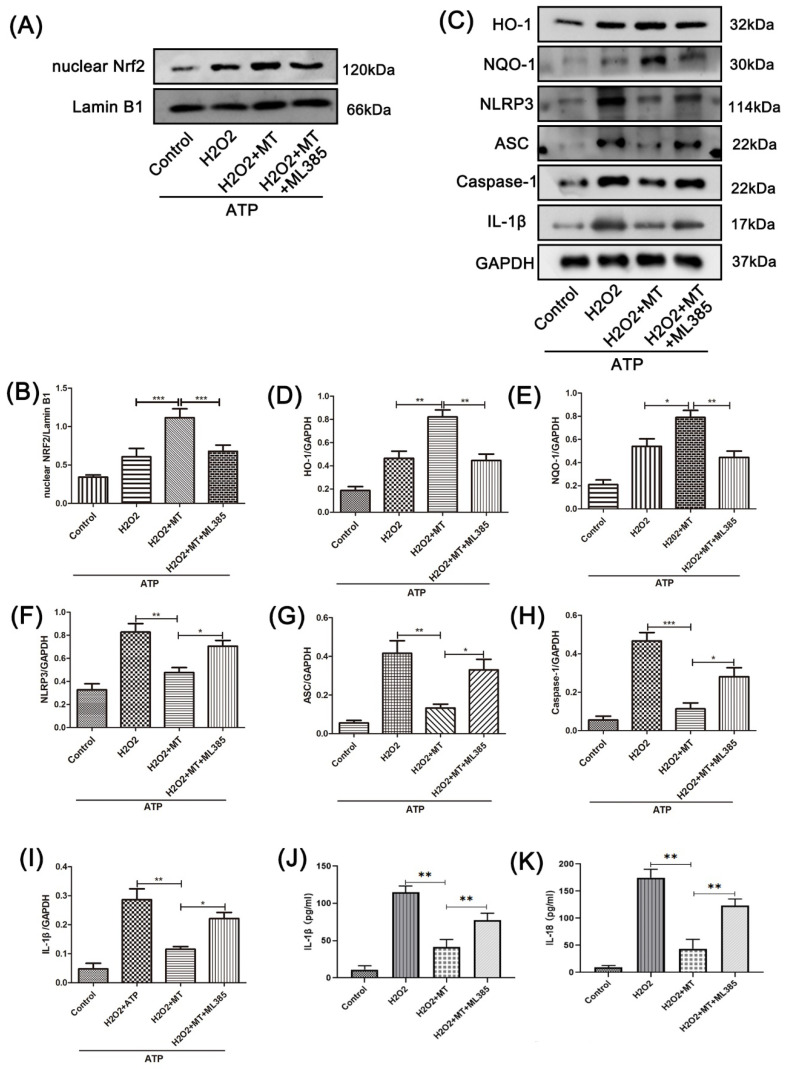
Melatonin suppresses the NLRP3 inflammasome in PC12 cells treated with H2O2 by activating the Nrf2/ARE pathway. (**A**–**I**) Western blot detection and statistical analysis of NLRP3-inflammasome-related proteins NLRP3, caspase-1, ASC, and IL-1β; and antioxidant-stress-related proteins nuclear Nrf2, HO-1, and NQO-1 in PC12 cells from each group (n = 6). (**J**,**K**) The levels of IL-1β and IL-18 in the supernatant of PC12 cells were determined by ELISA kits (n = 6). Data are shown as the means ± SD; * *p* < 0.05, ** *p* < 0.01, *** *p* < 0.001.

**Figure 8 cells-11-02809-f008:**
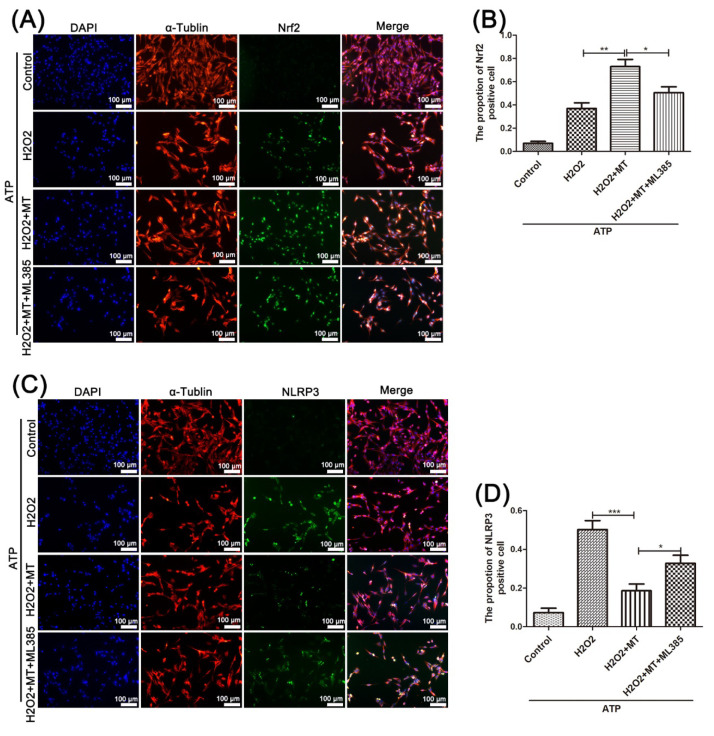
Immunofluorescence double staining of Nrf2 and NLRP3 in PC12 cells treated with H2O2. Immunofluorescence double staining was performed to test the protein level of Nrf2 (**A**,**B**) and NLRP3 (**C**,**D**) in PC12 cells from each group (n = 8, scale bar = 100 µm). Data are shown as the means ± SD; * *p* < 0.05, ** *p* < 0.01, *** *p* < 0.001.

## Data Availability

The datasets used and/or analyzed during the present study are available from the corresponding author on reasonable request.

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
