# Peer review of "Melatonin Attenuates Spinal Cord Injury in Mice by Activating the Nrf2/ARE Signaling Pathway to Inhibit the NLRP3 Inflammasome"

_cells, 2022, doi:10.3390/cells11182809_

Round 1

Reviewer 1 Report (Previous Reviewer 2)

The authors addressed most of my previous comments. 

Author Response

Thanks for your reviewing of our paper, if you have any question, please don't hesitate to contact us.

Reviewer 2 Report (New Reviewer)

This study investigated the effect of melatonin on the expression/activation of the NLRP3 inflammasome by activating the Nrf2/ARE signaling pathway in Spinal cord injury in-vivo and in-vitro.

There are a few points that need to be addressed:

Comments:

•             In the method section, a better explanation of JC-1 staining is required. How were monomers and aggregates observed (in which wavelength?), and how was the relative mitochondrial membrane potential calculated?

•             Figure 1B should be cited in the text. 

•             In Figure 1G, mitochondrial vacuolization should be indicated by arrows.

•             In Figure 2, the scale bars are not visible. If the same scale bar was used for all the images, it could be added to the figure legend. If it is different for different experiments, better to make it more visible.

•             Authors showed the effect of melatonin on NLRP3-related pathways using western-blot and immunofluorescence double staining. Although this approach gives information on the impact of melatonin on the protein expression of these genes as monomers, it does not show the NLRP3 activation, which requires the formation of the NLRP3 protein complex. In addition, although the expression of pro-IL-1β decreases, this information is insufficient to see the effect of melatonin on the maturation and secretion of already existing pro-IL-1β. An ELISA experiment for the release of mature IL-1β and IL-18 in an in-vitro experiment set up using PC12 cells would show a sign of NLRP3 activation status after melatonin treatment.

•             In Figure 3E; it the western-blot bands of Caspase-1 refers to caspase-1 or pro-caspase-1? It would be better to show both of them to see whether melatonin affects the protein expression of Pro/active caspase-1 or if it affects the maturation.

•             For all western blot images, the molecular weight of each protein should be added near the membrane.

•             In most in-vitro NLRP3 inflammasome studies, the LPS+ATP combination is used to activate NLRP3. However, in this study, to promote ROS, the authors used ATP following H2O2. A better explanation of the relation between H2O2 and NLRP3 activation will make the experiment workflow clearer. This explanation can be in the method, in the results part, in the legend of figure 7, or the discussion, depending on the author's preferences.

•             To evaluate the effect of melatonin on NLRP3 expression, using LPS primed and ATP treated PC12 cells is recommended to confirm the current findings.

•             Authors investigated the effect of melatonin on the rate of apoptosis in-vivo. However, in spinal cord injury, not only apoptosis but also autophagic and pyroptotic cell death could be observed. NLRP3 pathway is mainly linked to inflammatory cell death, pyroptosis and autophagy. Also, not only apoptotic cells, pyroptotic cells generate a low, positive signal when analyzed with a TUNEL assay. Therefore, stating the effect of melatonin-mediated NLRP3 suppression on programmed cell death as a general statement, rather than focusing only on apoptosis, would be more accurate.

Author Response

Please see the attachment。

Round 2

Reviewer 2 Report (New Reviewer)

Thank you to the authors for their sufficient responses. 

This manuscript is a resubmission of an earlier submission. The following is a list of the peer review reports and author responses from that submission.

Round 1

Reviewer 1 Report

This paper is not of high enough scientific quality to do a meaningful review.   All of the morphological and immunocytochemical studies of spinal cord tissue are seriously flawed because no information is provided about where images were taken with respect to the injury epicenter.  The extent of cell damage varies as a function of distance from the spinal cord lesion, so without information on where the images were taken, data are uninterpretable.   In my opinion, this paper should be editorially rejected.

Reviewer 2 Report

The manuscript investigates the role of melatonin in recovery after spinal cord injury, more specifically its effects agains neuroinflammation and secundary cell death upon injury. Their results indicate that maltonin acts by reducing neuronal mitochondrial dysfunction, the level of ROS and malondialdehyde, and enhancing the activity of superoxide dismutase and production of glutathione peroxidase. Mechanically, melatonin inhibits the activation of NLRP3 29 inflammasomes and reduces the secretion of pro-inflammatory factors by activating the Nrf2/ARE 30 signaling pathway.

The results are novel and deserve publication, but there are several serious problems with the manuscript that have to be addressed. First and foremost, the quality of English language is poor and the manuscript, including abstract is ridden with some very odd phrasing, e.g. "melatonin is capable to produce anti-oxidation and anti-inflammation effects"; "In the meantime, it also causes a neuroprotective effect.."; "the mechanism of melatonin in inflammatory regulation during SCI is poorly understood" etc, some of them being also scientifically wrong like "improving the recovery of nerve function after SCI" - the function of nerves is not affected by injury. 

Intorduction: line 57-58 the authors mention studies that show "that melatonin has a certain effect on neuroprotection" without citing them. It is impossible to judge from the introduction what was actually publihed about melatonin in neuroregeneration. 

The authors state that they have worked on the in vivo and in vivo spinal cord injury models. As far as i can see, the in vivo part is in M&Ms described unred "2.2. Surgical Preparations". I fail, however, to see how are the experiments on neuronal cell line "in vitro model of injury"? The recovery was followed for 28 days only and the values did not reach plateau in either group. Thus, better BMS scores on 14, 21 and 28 days do not guarantee that this trend would remain at further time-points. The nrf2 staining of the tissue does not look particularly specific. It does not help the authors case that the figure is described "Immunofluorescence double staining result of Nrf2 and NLRP3 protein in spinal anterior horn motor neuron cells of mice 3 days after spinal cord injury." It is a long confusing sentence. The same goes for HO-1 staining.

All in all, the manuscript has some potentially interesting results, but authors should put more effort in organizing their data in a presentable form. 

Reviewer 3 Report

In their manuscript entitled “Melatonin attenuates spinal cord injury in mice by activating Nrf2 / ARE signaling pathway to inhibit the NLRP3 inflammasome”, the authors studied the effect of melatonin on NLRP3 inflammasome activation and its signaling pathway (Nrf2/ARE), mitochondrial dysfunction and nerve function recovery following SCI. This is an interesting study, well-built and with interesting and clear results. The effect of Melatonin on spinal neurotoxicity and NLRP3 inflammasome has been published before (5 publications in PubMed form 2019, as for example: PMID: 30257055 and PMID: 35065245) but this a recent concept which needs more studies to confirm its benefit in SCI. The manuscript mostly suffers from several aspects that should be improved before publication. I hope my comment below help to improve the manuscript.

Introduction:

The introduction is short and does not provide clear explanation on the reason why the authors used both ex-vivo and in-vivo experiments. What the former brings and the complementarity. Similarly, the focus on mitochondrial dysfunction and Nrf2/ARE pathway’s anti-inflammation needs more in-depth explanation.

Methods:

Line 77: group melatonin+ML385: is presented without explanation. Inhibitor I guess?

Line 83: why a moderate contusion has been done? this is not explained. Do the authors think that similar results would be found with a more severe contusion? that shuold be suggested as a limitation or perspective of the study

Line 95: any references for the dose of Melatonin used?

Line 142: Timing for analyze is not explained either. Why 3 days would be the best time?

Results

Line 237: “14th day” after melatonin treatment? A paragraph at the beginning of the methods to explain the study design would be helpful. How many animals were used for 3-, 14- or 28 days experiments?

Again, the complementarity between in vivo and ex-vivo is not clear. Why doing the same experiment both in vivo and in PC12 cells?

Discussion/Conclusion:

Line 392: “disease” is not a good word for SCI. please change. 

A paragraph on clinical impact and perspective is missing. What is the translational aspect in this study? Where are we regarding to clinical using of melatonin? What are future directions?